# Red Wine Consumption and Cardiovascular Health

**DOI:** 10.3390/molecules24193626

**Published:** 2019-10-08

**Authors:** Luigi Castaldo, Alfonso Narváez, Luana Izzo, Giulia Graziani, Anna Gaspari, Giovanni Di Minno, Alberto Ritieni

**Affiliations:** 1Department of Pharmacy, Faculty of Pharmacy, University of Naples “Federico II”, Via Domenico Montesano 49, 80131 Naples, Italy; luigi.castaldo2@unina.it (L.C.); alfonsonsimon@gmail.com (A.N.); luana.izzo@unina.it (L.I.); giulia.graziani@unina.it (G.G.); annagaspari@virgilio.it (A.G.); 2Department of Clinical Medicine and Surgery, University of Naples “Federico II”, Via S. Pansini 5, 80131 Naples, Italy; diminno@unina.it

**Keywords:** red wine, resveratrol, polyphenols, alcohol, cardioprotective, antioxidants

## Abstract

Wine is a popular alcoholic beverage that has been consumed for hundreds of years. Benefits from moderate alcohol consumption have been widely supported by the scientific literature and, in this line, red wine intake has been related to a lesser risk for coronary heart disease (CHD). Experimental studies and meta-analyses have mainly attributed this outcome to the presence in red wine of a great variety of polyphenolic compounds such as resveratrol, catechin, epicatechin, quercetin, and anthocyanin. Resveratrol is considered the most effective wine compound with respect to the prevention of CHD because of its antioxidant properties. The mechanisms responsible for its putative cardioprotective effects would include changes in lipid profiles, reduction of insulin resistance, and decrease in oxidative stress of low-density lipoprotein cholesterol (LDL-C). The aim of this review is to summarize the accumulated evidence correlating moderate red wine consumption with prevention of CHD by focusing on the different mechanisms underlying this relationship. Furthermore, the chemistry of wine as well as chemical factors that influence the composition of the bioactive components of red wine are also discussed.

## 1. Introduction

Coronary heart disease (CHD) and stroke are the leading causes of mortality, disability, and death in developed countries [1]. Most CHDs are due to atherosclerosis, a degenerative process of the arteries which is triggered by oxidative stress and chronic inflammatory status [2,3]. Smoking, arterial hypertension, hypercholesterolemia, diabetes mellitus, overweight/obesity, lack of physical activity, and genetic factors are known to play a role in determining cardiovascular risk [4].

Although excessive alcohol intake is associated with the development of chronic diseases and other serious problems, a wealth of data from scientific evidence support an inverse relationship between moderate alcohol consumption and the risk of CHD [5]. Moderate alcohol consumption is defined in the Dietary Guidelines for Americans 2015–2020 as up to one unit of alcohol per day for women and up to two units of alcohol per day for men [6].

Several studies provide evidence that light–moderate alcohol consumption is associated with a higher level of high-density lipoprotein cholesterol (HDL-C), a lower incidence of type-2 diabetes (T2D), and a reduction of lipid oxidative stress [7,8,9,10]. Such epidemiological studies have supported that red wine consumption is more CHD-preventative in comparison to the intake of other alcoholic beverages [11]. It is uncertain whether the apparent beneficial properties for health attributed to the consumption of red wine are due solely to the presence of alcohol or also to the concerted action of alcohol and antioxidant compounds other than alcohol present in red wine [12]. In addition to alcohol, red wine contains a wide range of active compounds—polyphenols—with antioxidant and anti-inflammatory properties that could contribute to protection from atherosclerotic pathologies [13].

Light–moderate drinking of red wine has been proposed as a possible explanation for the epidemiological phenomenon known as the “French Paradox” [14], which indicates that the French population shows relatively lower CHD incidence/mortality rate compared with other Western populations, despite their diets contain higher amounts of total fat and saturated fatty acids.

The main objective of this review is to summarize the various red wine components and their cardioprotective potential. In addition, we discuss the putative mechanisms and the chemical factors that influence the activity of the bioactive components of red wine. All studies regarding the relationship between red wine consumption and CHD published over the last decade have been taken into consideration.

## 2. Bioactive Components in Red Wine

Red wine contains high concentrations of polyphenolic compounds such as flavonoids (catechin, epicatechin, quercetin, anthocyanins, and procyanidins), resveratrol (3,5,4′-trihydroxystilbene), and polymeric tannins [15]. In general, red wine is rich in polyphenols and may be considered as an important polyphenols source in the diet [16]. The presence of phenolic compounds in red wine seems to be crucial, since scientific studies have reported that these important secondary metabolites are responsible for desirable biological actions, including cardiovascular protection effects [17,18,19].

### 2.1. Non-Flavonoid

Non-flavonoid phenolic components of grapes and wine include three main groups: stilbenes, benzoic acids, and cinnamic acids [20]. The last two groups can be present as hydroxybenzoic and hydroxycinnamic acids. Benzoic acids are present in the grape as well as in oak wood and, during storage, can migrate into wine [21]. In general, this group of compounds is present in red wine at concentrations ranging from 60 to 566 mg/L.

#### 2.1.1. Hydroxybenzoic Acids

Hydroxybenzoic acids (HBAs) are phenolic metabolites with a general C6–C1 structure. In red wine, the most abundant HBAs are represented by *p*-hydroxybenzoic, gallic, vanillic, gentisic, syringic, salicylic, and protocatechuic acids [22,23]. As reported, the different hydroxybenzoic acids may occur mainly in their free forms [24]. Gallic acid is an important HBA present in red wine but not in grape and it is probably formed by hydrolysis of tannins (condensed or hydrolyzable) during the vesting period in oak wood [25]. The total amount of hydroxybenzoic acids in red wine is expected to range from undetectable to 218 mg/L, as shown in Table 1.

#### 2.1.2. Hydroxycinnamic Acids

Hydroxycinnamic acids are the major phenols in both grapes and wine [26,27]. Caffeic, coumaric, and ferulic acids are some of the most important compounds in this polyphenol sub-class [28]. Natural hydroxycinnamic acids are not found in grape, appearing as their tartaric acid esters or diesters [29]. The main hydroxycinnamic acids of wine are *p*-coutaric, caftaric, and fertaric acids [30]. In nature, hydroxycinnamic acids exist in two isomeric forms, but the *trans*-form is the most abundant in both grapes and wine [31]. Coutaric acid is mainly contained in the grape skin, while *trans*-caftaric and *trans*-fertaric acid are mainly present in the pulp [32]. The amount of hydroxycinnamic acids in different red wines was found to range from 60 to 334 mg/L, as shown in Table 1.

#### 2.1.3. Resveratrol

Resveratrol (3, 4, 5 trihydroxystilbene), a non-flavonoid polyphenolic compound, is a common phytoalexin synthesized in response to the attack of bacteria and fungi [33]. It is present in more than 70 plant species, including berries, peanuts, cocoa, and grape skin [34]. Resveratrol has two phenol rings linked to each other by a styrene double bond in its chemical structure [35]. It exists as *cis*- (Z) and *trans*- (E) isomers, and both have been detected in wine at variable concentrations [36,37] ranging from 0.1 to 7 mg/L and from 0.7 to 6.5 mg/L, respectively [38]. Variability is mainly due to grape cultivar, geographic origin, oenological practices, and wine type [39].

Several studies regarding the health benefits of *trans*-resveratrol in maintaining human health and preventing a wide variety of human diseases are available [40,41,42,43,44,45].

Magyar et al. [46] investigated the cardioprotective effect of low doses of resveratrol (10 mg/day) in 40 patients with stable coronary artery disease. The results showed that resveratrol intake displayed a significantly lowered low-density lipoprotein-cholesterol (LDL-C) level, improved endothelial function and left ventricular diastolic function, and protected against some unfavorable hemorheological changes.

Romain et al. [47] investigated the benefits of a grapevine-shoot phenolic extract (Vineatrol 30) that contained considerable amounts of resveratrol (about 15.2%) on the cardiovascular system in hamsters fed a high-fat diet. The results showed that Vineatrol 30 was able to prevent aortic fatty streak deposition by increasing antioxidant and anti-inflammatory activities.

Fujitaka et al. [48] investigated the effects of a high dose (100 mg/day) of modified resveratrol, Longevinex, on the metabolic profile, inflammatory response, and endothelial function in subjects with metabolic syndrome (MetS). The results showed that after three weeks, modified resveratrol specifically improved the endothelial function in patients with MetS.

D’Archivio et al. [49] highlighted the possible influence of the matrix sugar content on resveratrol bioavailability, since bioavailability appears to be higher for the aglycone form in comparison to its glycosides in grape juice.

Wang et al. [50] demonstrated that *trans*-resveratrol is rapidly metabolized by glucuronidation and/or sulfation reactions as well as by hydrogenation of the aliphatic double bond, probably mediated by intestinal bacterial metabolism.

Regarding its bioactivity, *trans*-resveratrol may represent a promising dietary supplement and is currently proposed as a therapeutic agent for many diseases [51,52,53,54,55].

### 2.2. Flavonoids

Flavonoids are plant-derived phytochemicals with antioxidant properties that account for over 85% of the phenolic components in red wine [56]. Flavonoids share a common basic structure consisting of a three-ring system with a central oxygen-containing ring (C ring) [57]. The substitution of the central pyran ring and the different oxidation degree are responsible for their chemical diversity [58]. On the basis of these differences, the flavonoids comprise a wide range of compounds such as flavones, flavonols, flavanols, anthocyanidins, and anthocyanins [59]. Natural flavonoids can exist in their free form (aglycone) or as glycosides condensed with the hydroxyl group of a sugar such as glucose, galactose, rhamnose, glucuronide, xylose, and arabinose [60]. They are widely distributed primarily in vegetables, seeds, nuts, spices, herbs, cocoa, and grape skin. The total level of flavonoids can vary from 150 mg/L to 650 mg/L

Over the last decade, a large amount of experimental and epidemiological investigations has supported the protective effect of flavonoids on cardiovascular and chronic degenerative diseases [61,62]. The cardioprotective effects ascribed to flavonoids against atherosclerosis development might be due to the ability of flavonoids to improve the lipid profiles and reduce insulin resistance and oxidative stress, especially of LDL-C, as suggested by several studies [63,64,65,66].

#### 2.2.1. Flavones

Flavones display three functional groups i.e., hydroxy and carbonyl groups and conjugated double bonds between C2 and C3 in the flavonoid skeleton. These compounds were found in grape skin and wine in both aglycones and glycosides forms. In grapes, luteolin is the only flavone, present in levels ranging from 0.2 to 1 mg/L [67,68,69]. Flavones are known to have a wide range of important biological properties including antioxidant, anti-inflammatory, and anti-tumor activities and are also used as supplements in the treatment of CHD and neurodegenerative disorders [70].

Li et al. [71] investigated the cardioprotective properties of total flavone administrations of *Choerospondias axillaries* (from 75.0 to 300.0 mg/kg) in a rat model of ischemia–reperfusion (I/R). The results indicated that flavones intake was able to reduce heart pathologic lesions and improve the cardiac function by increasing antioxidative activities.

#### 2.2.2. Flavan-3-ols

Flavanols are benzopyrans that include simple monomeric and polymeric forms which are contained in noteworthy concentrations in red wine. The most important flavanols in grape are catechin and its enantiomer epicatechin, biosynthetic precursors of proanthocyanidins, which are responsible for the structure and astringency of wine [56]. The catechin and epicatechin levels in red wine were reported to range from −50 to 120 mg/L [32,68,69]. Moreover, catechin levels to 1000 mg/L were recorded especially in selected old red wines [49].

Several studies indicate that flavan-3-ols may exert cardioprotective actions, which might be due to the ability of flavan-3-ols to increase nitric oxide (NO) bioactivity and decrease superoxide production [72]. Ramirez-Sanchez et al. [73] studied the possible stimulation of endothelial nitric oxide synthase (eNOS) by epicatechin, an enzyme that generates the vasoprotective molecule NO in human coronary endothelial cells. The results showed that acute administration of epicatechin may induce eNOS activation in endothelial cells.

#### 2.2.3. Flavonols

Flavonols are often characterized by a hydroxyl group in C3 (3-hydroxyflavones), thus being often named 3-hydroxyflavones. Flavonols found in red wine include aglycons such as myricetin, quercetin, kaempferol, and rutin and their respective glycosides which can be glucosides, glucuronides, galactosides, and diglycosides. These main flavonols can be found at a total concentration ranging from 12.7 to 130 mg/L [67,68,69]. These compounds are known to play a wide range of biological activities and are considered the main active compounds within the flavonoids group [74].

Annapurna et al. [75] investigated the cardioprotective actions of quercetin and rutin (from 5 to 10 mg/kg) in both normal and diabetic rats. Quercetin and rutin intake showed a protective effect in I/R-induced myocardial infarction in normal as well as diabetic rats. Therefore, it was concluded that quercetin and rutin protection could be due to an increased antioxidant activity.

#### 2.2.4. Anthocyanins

Anthocyanin is the glycosylated form of the so-called anthocyanidin. The general molecular structure of anthocyanins is based on the flavilium or 2-phenylbenzopyrilium cation, with hydroxyl and methoxyl groups are present at different positions of the basic structure. The great variety of anthocyanins found in nature is defined by the number and position of hydroxylated groups and the number and the position of conjugated sugar and acyl moieties in their structure [32]. Anthocyanins, namely, malvidin, cyanidin, delphinidin, petunidin, peonidin, and pelargonidin, have been detected in both grape and red wine at levels ranging from 90 to 400 ng/mL [76,77,78]. Anthocyanins are usually found in their glucosyde forms, but rhamnose, xylose, and galactose have also been observed as common sugar moieties. Anthocyanins are also found with acyl substituents bound to sugars, aliphatic acids, and cinnamic acids. 

An ever-expanding amount of scientific evidence supports a protective role of habitual dietary intake of anthocyanins against age-related chronic conditions including CHD [79,80,81,82,83].

McCullough et al. [84] investigated the association between flavonoid intake and risk of death from CHD in a cohort study of 98,469 participants. The results confirmed that anthocyanidins and proanthocyanidins are associated with lower CHD risk, concluding that food sources rich in flavonoids should be considered for CHD risk reduction.

Similarly, Cassidy et al. [85] evaluated the relationship between intake of different flavonoid classes and multiple inflammatory biomarkers assessed in combination by an inflammation score (IS) in a cohort study of 2375 participants. The results showed that higher intakes of anthocyanins were inversely associated with reductions of the IS score (73%).

Huang et al. [86] studied the most common red wine anthocyanins, malvidin-3-glucoside and malvidin-3-galactoside, to evaluate their effect on the inflammatory response in endothelial cells. An anti-inflammatory effect was shown by both malvidin-3-glucoside and malvidin-3-galactoside, and a synergistic effect of the two compounds was also found.

Anthocyanins represent promising molecules for the development of therapeutic agents to prevent chronic inflammation in many diseases.

#### 2.2.5. Tannins

Tannins are another important subgroup of phenols present in red wine that contribute to astringency and are also implicated in reactions that lead to browning, especially in white wines. They can be classified into two main classes, namely, hydrolyzable and condensed tannins. The latter forms, polymers of flavan-3-ol without sugar residues, are predominant in grape and wine, while hydrolyzable tannins are naturally present in oak barrels. The total content of tannins ranges from 1.1 to 3.4 g/L [78,87].

In vivo studies carried out in animals and humans suggest that tannins possess potent antioxidant, anti-inflammatory, and radical scavenging activity able to promote benefits to human health [88,89].

#### 2.2.6. Hydrolyzable tannins

The basic unit of hydrolyzable tannins are represented by gallic and ellagic acids usually esterified with glucose or related sugars. They are more susceptible to hydrolysis than condensed tannins induced by pH changes and enzymatic or non-enzymatic processes. On the basis of the type of phenolic acid present in their structure, hydrolyzable tannins can be divided in gallotannins and ellagitannins, usually found as a mixture in plant sources. The hydrolyzable tannins are not found in grapes but are extracted from barrels wood during wine aging and are thus proposed in the literature as a marker of maturity for this type of wine. As a result of differences in the aging process and type of wood, the final content of hydrolyzable tannins can strongly vary from 0.4 to 50 mg/L [78,90,91,92,93].

#### 2.2.7. Condensed tannins

Condensed tannins (or proanthocyanidins) are oligomer flavonoids that contribute to astringency in wine. The depolymerization of the condensed tannins under oxidative conditions leads to the formation of proanthocyanidins. Flavan-3-ols and their precursor flavan-3,4-diols (leucoanthocyanidin) are the main components of condensed tannins in nature and can be found at concentrations levels from 1.2 to 3.3 g/L [78,87].

## 3. Factors Influencing Bioactive Compounds and Composition of Wine

The genetic factors (variety) of grapes and the vinification conditions are considered the main factors that influence the wine polyphenolic composition [32]. Moreover, some studies have shown that other different variables can also act on grapes phenolic accumulation [97,98]. The agroecological factors that mostly influence the quali–quantitative polyphenol content of the grapes can be summarized in the geographic origin of grapes, the climatic and soil conditions, the exposure to diseases, and the degree of ripeness [99]. In red wine production, the methods of winemaking (maceration, fermentation, clarification, aging, etc.) and the processing operations (ionic exchange, filtration, centrifugation) can modify significantly the composition and the concentration of phenolic compounds [100,101]. Moreover, during wine maturation and aging, the concentration of monomeric phenols present in wine declines constantly, while complex and stable molecules derived from the condensation of catechins, anthocyanins, and proanthocyanidin are formed [101,102]. Consequently, the polyphenolic composition of grapes differs from that of their corresponding wines. Some of the reactions occurring during the winemaking process are enzymatic oxidation, electrophilic substitution, complexation, and hydrolysis [103,104]. Moreover, new polyphenolic compounds may also be present in wine for environmental reasons like aging in oak barrels, which promotes the extraction of low-molecular-weight phenolic compounds such as flavonoids and of hydrolyzable tannins, modifying the organoleptic characteristics as well as the health impact of a wine [105].

## 4. Putative Mechanisms of Action

A large number of epidemiological studies and meta-analysis have consistently shown that light–moderate drinking of red wine has a protective effect against CHD [106,107]. Several plausible underlying biological mechanisms have been postulated to explain the beneficial effects of light–moderate red wine consumption as well as of the phenolic compounds contained in red wine on the development of CHD and atherosclerosis [108,109]. Understanding the mechanisms by which light–moderate drinking of red wine improves the cardiovascular function is crucial for the treatment and prevention of CHD.

### 4.1. Lipid Profile

Epidemiological studies have consistently shown associations between hyperlipidemia and risk of developing CHD, obesity, and T2D. Light–moderate drinking of alcohol, especially red wine, is associated with beneficial changes in lipid homeostasis, as shown by the results of several clinical trials and meta-analyses.

Da Luz et al. [110] evaluated the association between moderate red wine consumption and changes in HDL-C levels and in the coronary vasculature. The study included 205 subjects (101 and 104 drinkers and abstainers, respectively) aged around 60 years. Red wine drinkers displayed an HDL-C level significantly higher than the abstainers and a protective effect on coronary lesions.

Marques-Vidal et al. [111] had similar results in a large cohort. The study included 5409 subjects categorized as abstainers (0 drinks/week, *n* = 1463), moderate alcohol drinkers (1–13 drinks/week, *n* = 2972), high alcohol drinkers (14–34 drinks/week, *n* = 867), and very high alcohol drinkers (≥35 drinks/week, *n* = 107). The results showed that alcohol consumption increased HDL-C levels rather than polyphenols in light moderate drinkers and partly explained the cardioprotective effect displayed by alcohol consumption.

These results were corroborated by Park et al., [112] who investigated the benefits of moderate consumption of alcohol in a hypertensive population with a focus on the lipid profile. The study included 2014 participants aged 20–69 years. The results showed that alcohol consumption was negatively associated with prevalence of low HDL-C, whereas the amount of triglycerides increased with a higher alcohol intake.

Magnus et al. [113] investigated the hypothesis that moderate alcohol intake exerts its cardioprotective function by increasing HDL-C levels with a cohort study of 149,729 participants. The results showed that increasing HDL-C levels is not a relevant mechanism by which ethanol exerts its cardioprotective effect.

A recent meta-analysis [114] examined the effect of moderate alcohol consumption on lipid profile, concluding that alcohol consumption significantly increased the levels of HDL-C, apolipoprotein A1, and adiponectin. Moreover, the results showed that alcohol did not significantly changed triglycerides levels.

The findings above-reported support an increase in the plasma HDL-C concentration level as a result of chronic, moderate alcohol consumption. Higher HDL levels have been consistently observed in cohort studies regarding alcohol consumption and attributed to alcohol itself. In fact, alcohol, rather than polyphenols, appears to be responsible for the increase of plasma HDL in wine light–moderate drinkers. Table 2 summarizes these studies. The studies that showed positive changes of lipid metabolism, except for HDL-C, as a consequence of light–moderate drinking of alcohol were inconclusive, especially cross-sectional studies where some outcome had a longer half-life time than those analyzed [115]. On the other hand, the effects of light–moderate drinking of alcohol, including red wine, on triglycerides, LDL, very low density lipoproteins (VLDL), and lipoprotein (a) are unclear and still under debate [116].

### 4.2. Glucose Metabolism

The cardioprotective effect of red wine consumption may partly be explained by the association between moderate red wine consumption and a lower incidence of T2D.

Chiva-Blanch et al. [115] showed that moderate consumption of red wine (30 g of alcohol per day) and dealcoholized red wine decreased the homeostasis model assessment of insulin resistance values (HOMA-IR) and plasma insulin after 4 weeks in 67 men at high cardiovascular risk. These results suggest that the beneficial effects could be mediated by antioxidant compounds present in red wine, while alcohol did not seem fundamental to obtain such effects.

Brasnyó et al. [117] investigated the effects of low doses of resveratrol (2 × 5 mg/day) on glucose metabolism in 19 T2D patients. After 4 weeks, resveratrol improved insulin resistance and increased the phosphorylation of protein kinase B (AKT), which plays a key role in insulin signaling by interfering directly with glycogen synthesis. Therefore, it was concluded that resveratrol might be used for medicinal application.

Da Luz et al. [118] evaluated the association of moderate red wine consumption with changes in glucose levels and diabetes. The study included 205 subjects (101 and 104 drinkers and abstainers, respectively) aged around 60 years. Red wine drinkers displayed a significantly lower incidence of diabetes and lower glucose levels compared to abstainers.

A recent meta-analysis [119] of 20 cohort studies comprising 477,200 subjects confirmed the U-shaped relationship between moderate amounts of alcohol consumption and risk of incident T2D for both sexes compared with lifetime abstainers. The amount of alcohol that showed higher protective effects was 22 g/day for men and 24 g/day for woman, while over 60 and 50 g/day of alcohol were deleterious for men and women, respectively. Therefore, in this study, the amount of polyphenols was not considered, and the protective effect was attributed to alcohol.

The cardioprotective effects of moderate alcohol consumption were corroborated by Mekary et al. [120] through a large prospective study including 81,827 participants on the impact of alcohol consumption and the positive association between glycemic load (GL) and the incidence of T2D. They found that a high alcohol intake (≥15 g/day) attenuated the effect of GL on T2D incidence.

Ramadori et al. [121] conducted a study on diet-induced obese and diabetic mice to evaluate the impact of approximately 79.2 ng/day intracerebroventricular infusion of resveratrol on glucose metabolism. The results showed a normalized hyperglycemia and improved hyperinsulinemia by the activation of SIRT 1 expressed in the brain. Table 3 summarizes these studies.

These findings suggest that a light to moderate alcohol consumption, especially with red wine, may be associated with improved insulin resistance and with a lower incidence of diabetes, providing another potential explanation for the reduction of cardiovascular events associated with moderate alcohol intake.

### 4.3. Oxidative Stress

Many important cardioprotective effects of wine polyphenols can be attributed to their capacity to react with reactive nitrogen species (RNS) or to interfere with RNS production. Wine polyphenols are well recognized as potent antioxidant compounds and radical scavengers of peroxynitrite, a reactive substance produced by the reaction between NO and the superoxide anion [122,123]. The inverse association between red wine consumption and mortality from cardiovascular diseases may be explained by the capacity of red wine polyphenols to reduce LDL oxidation [124]. These findings showed that the beneficial effects on LDL oxidation could be exerted by a higher antioxidant activity of red wine compared to beverages with no polyphenolic content.

Estruch et al. [125] studied the benefits of moderate consumption of red wine compared to gin, an alcoholic beverage without polyphenolic content, on the lag phase time of LDL particles. The study was conducted with 40 healthy men aged 38 years, concluding that after 28 days of moderate consumption of red wine (30 g/day). Compared to gin, red wine intake increased up to 11.0 min the lag phase time of LDL oxidation, probably due to its high polyphenolic content. 

Similarly, Chiva et al. [126] checked the effects of alcoholic and dealcoholized red wine and gin intake on plasma NO and blood pressure in 67 subjects at high cardiovascular risk. After 4 weeks, the results showed that dealcoholized red wine was able to decrease systolic and diastolic blood pressure and increase plasma NO concentration.

Egert et al. [127] evaluated changes in markers of oxidative stress following quercetin intake in 93 overweight or obese subjects aged 25–65 years. Quercetin is an important flavonoid present in high amounts in red wine and grapes. After 6 weeks, 150 mg/day of quercetin supplementation significantly decreased the plasma concentrations of oxidized LDL. Therefore, it was concluded that quercetin may provide protection against CHD.

Bulut et al. [128] evaluated the effects of alcoholic (red wine and liquor) and non-alcoholic (mineral water and Coke) beverages consumed during a high-fat meal once a week for 4 weeks on circulating microparticles (MPs) in 10 healthy males. Volunteers in the red wine and liquor groups consumed the same amount of alcohol. The results indicated that the number of MPs increased after a single high-fat meal (increase by about 62%), but red wine consumption decreased these negative effects (increase by about 5%). Table 4 summarizes these studies.

These findings support that moderate red wine consumption may act as an antioxidant by decreasing oxidized LDL plasma levels and increasing plasma NO concentration. Scientific evidence indicates that oxidized LDL may play a major role in the onset and progression of oxidative stress-associated diseases, such as atherosclerosis [129,130]. Moreover, increased oxidized LDL plasma levels were predictive of future myocardial infarction [131]. Nevertheless, the beneficial effects of moderate red wine consumption on LDL oxidation seem to be independent of its alcohol component.

## 5. Conclusions

In the last decades, several human and animal studies have indicated that moderate red wine consumption has beneficial effects on health. Phenolic compounds present in red wine have shown antioxidant and anti-inflammatory properties, being able to reduce insulin resistance and to exert a beneficial effect by decreasing oxidative stress. As a consequence, a clear effect on the reduction of risk factors and the prevention of cardiovascular diseases have been observed. Different mechanisms are involved in the cardioprotective effects of moderate red wine consumption: while alcohol appears to be responsible for increasing plasma HDL-C, the polyphenolic component may play a key role in the reduction of T2D incidence and LDL oxidation. In light of these considerations, a moderate intake of red wine may produce cardioprotective effects. However, more in-depth knowledge is required in order to understand the molecular basis of the potential mechanisms involved.

## Figures and Tables

**Table 1 molecules-24-03626-t001:** Main representative groups of polyphenols present in red wine [67,68,69,77,94,95,96].

Group	Subclass	Main Representatives	Range in mg/L	Characteristic Structure
Non-flavonoid				
	Hydroxybenzoic acids	Gallic, ellagic, parahydroxybenzoic, protocatechuic, vanillicand syringic acids	0–218.0	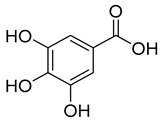 Gallic acid
	Hydroxycinnamic acids	Coutaric, caftaric, and fertaric acids	60.0–334.0	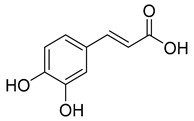 Caffeic acid
	Stilbenes	Resveratrol	0.1–7.0	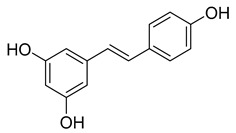 Resveratrol
Flavonoids				
	Flavones	Luteolin	0.2–1.0	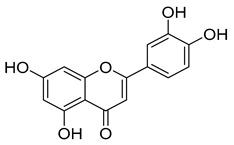 Luteolin
	Flavan-3-ols	Catechin and epicatechin	50.0–120.0	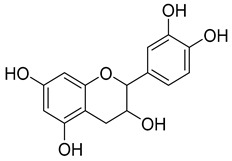 Catechin
	Flavonols	Myricetin, quercetin, kaempferol, and rutin	12.7–130.0	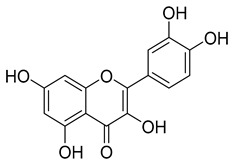 Quercetin
	Anthocyanins	Malvidin, cyanidin, peonidin, delphinidin, pelargonidin, petunidin	90.0–400.0	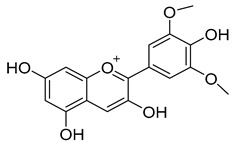 Malvidin

**Table 2 molecules-24-03626-t002:** Summary of studies assessing the relationship between moderate alcohol consumption and lipid profile. HDL-C: high-density lipoprotein-cholesterol.

Study Model	Outcome(s)	Study Characteristics	Main Findings	References
Drinkers vs. abstainers	HDL-C level and changes in the coronary vasculature	Benefits of moderate consumption of red wine	Drinkers displayed a significantly higher HDL-C level and a protective effect on coronary lesions	[110]
Men	HDL-C level	The study included 5409 subjects categorized in abstainers, moderate, high, and very high alcohol drinkers	Alcohol consumption increased HDL-C levels	[111]
Hypertensive population	Lipid profile	Benefits of moderate consumption of alcohol	Alcohol consumption was negatively associated with prevalence of low HDL-C levels, whereas the prevalence of high triglyceride levels increased with increasing amounts of alcohol intake	[112]
Cohort study	Cardioprotective function	Increasing HDL-C levels as the mechanism used by alcohol to exert its cardioprotective function	Increasing HDL-C levels is not a relevant mechanism by which ethanol exerts its cardioprotective effect	[113]
Meta-analysis	Lipid profile	Effect of moderate alcohol consumption on lipid profile	Alcohol consumption increased the levels of HDL-C, apolipoprotein A1, and adiponectin but not triglycerides levels.	[114]

**Table 3 molecules-24-03626-t003:** Summary of studies assessing the impact of red wine consumption on glucose metabolism. HOMA-IR: homeostasis model assessment of insulin resistance, T2D: type-2 diabetes, GL: glycemic load.

Study Model	Outcome (s)	Study Characteristics	Main Findings	References
Men	Glucose metabolism	Light–moderate alcohol consumption (red wine, dealcoholized red wine, and gin)	Dealcoholized red wine decreased plasma insulin and HOMA-IR values	[115]
Drinkers vs. abstainers	Glucose level and diabetes	Benefits of moderate consumption of red win	Drinkers showed a lower incidence of diabetes and lower glucose levels compared to abstainers	[110]
T2D	Glucose metabolism	Effect of resveratrol on glucose metabolism	Resveratrol improved insulin resistance and increased AKT phosphorylation	[117]
Meta-analysis	T2D incidence	Effect of moderate alcohol consumption on the incidence of T2D	Light–moderate alcohol consumption decreased the incidence of T2D	[119]
Cohort study	GL and incidence of T2D	Impact of alcohol consumption and positive association between GL and T2D	High alcohol intake (≥15 g/day) attenuates the effect of GL on T2D incidence	[120]
Obese and diabetic mice	Glucose metabolism	Impact of intracerebroventricular infusion of resveratrol on glucose metabolism	Normalized hyperglycemia and improved hyperinsulinemia mediated by activating SIRT 1 expressed in the brain	[121]

**Table 4 molecules-24-03626-t004:** Summary of studies assessing the impact of red wine consumption on oxidative stress. LDL: low-density lipoprotein.

Study Model	Outcome(s)	Study Characteristics	Main Findings	References
Men	Lag phase time of LDL particles	Benefits of moderate consumption of red wine, dealcoholized red wine, and gin.	Red wine consumption showed increased lag phase time of LDL oxidation up to 11.0 min	[125]
High cardiovascular risk	Plasma nitric oxide, systolic and diastolic pressure	Effects of alcoholic and dealcoholized red wine and gin on plasma NO and blood pressure	Dealcoholized red wine reduced systolic and diastolic blood pressure and increase plasma NO concentration	[126]
Overweight or obese subjects	Concentrations of oxidized LDL	Changes in markers of oxidative stress following 150 mg/day of quercetin supplementation	Quercetin significantly decreased plasma concentrations of oxidized LDL	[127]
Men	Circulating microparticles	Benefits of moderate consumption of red win, dealcoholized red wine and gin during a high-fat meal	Red wine consumption decreased circulating microparticles	[128]

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
