# Peer review of "Red Wine Consumption and Cardiovascular Health"

_molecules, 2019, doi:10.3390/molecules24193626_

Round 1
Reviewer 1 Report
The review summarize the literature concerning the moderate red wine consumption correlated to the prevention of CHD. The chemistry of wine as well as chemical factors that influence the bioactive components of red wine were also reported in this review.
General comments
The knowledge about 2.2.1. Flavones, 2.2. Flavanols and 2.2.3. Flavonols and 2.2.5. Tannins should be improved, there are no bibliographic reference on the 2.2.3. Flavonols, 2.2.5. Tannins and 2.2.6. Hydrolyzable tannins??? The Information that summarized the molecular mechanism in Lipid profile, Glucose metabolism, Oxidative should be better explain, in order to understand why red wine improves the cardiovascular functions. The conclusion needs to be improved. Explain the sentence “Nevertheless, alcohol appears to be responsible for the increase of plasma HDL-C rather than polyphenols in wine moderate drinkers.”Author Response
Response to Reviewer 1 Comments
Point 1: The knowledge about 2.2.1. Flavones, 2.2. Flavanols and 2.2.3. Flavonols and 2.2.5. Tannins should be improved, there are no bibliographic reference on the 2.2.3. Flavonols, 2.2.5. Tannins and 2.2.6. Hydrolyzable tannins???
Response 1: As suggested by the reviewer 1 the mentioned section has been improved and the changes, includes the references, are added at the manuscript.
Point 2: The Information that summarized the molecular mechanism in Lipid profile, Glucose metabolism, Oxidative should be better explain, in order to understand why red wine improves the cardiovascular functions.
Response 2: Following the indications given by reviewer the authors improved the mentioned sections in order to better explain the cardiovascular functions of red wine consumption.
Point 3: The conclusion needs to be improved. Explain the sentence “Nevertheless, alcohol appears to be responsible for the increase of plasma HDL-C rather than polyphenols in wine moderate drinkers.”
Response 3: As suggested by reviewer the sentence mentioned in the conclusion section has been clarified and reported as:
"Different mechanisms are involved regarding cardioprotective effects of moderate red wine consumption: alcohol appears to be responsible for increasing plasma HDL-C whereas the polyphenolic component may play a key role in the reduction of T2D incidence and LDL oxidation."
Authors thanks the reviewer comments!
Reviewer 2 Report
The overall idea of the manuscript is valid, the subject is relevant, but there already are plenty of works about this research field. This type of review is not new. The manuscript is well written and the English language is generally correct. The manuscript despite being small are too descriptive and should be rearranged in order to improve its quality. There are many aspects in which the manuscript should be ameliorated. If the authors agree to do these improvements, I recommend the acceptance of the manuscript to be published in Molecules with major revision:
- The abstract should be revised and the sentences shortened (e.g. lines 13/17);
- The term “antioxidant ingredients” should be revised and improved all along the manuscript;
- Correct “Trans-Resveratrol” at the beginning of the sentence;
- The methodology used for the bibliographic search and the criteria for the articles selection should appear at the end of the introduction, before the results;
- In section 2, sometimes the compounds levels are referred and other times not; as so, it should be uniform, the levels should always be referred;
- Insert reference in section 2.2.3;
- Taking the section “3. Factors influencing phenolic content and composition of wine”, the non-phenolic content is not influenced by external factors?”; authors should clarify this question and add this topic in the manuscript;
- The manuscript is very descriptive and should discuss more the ideas and the results of the studies cited: Authors should improve this aspect;
- What authors mean when they write "Principal studies". What was the criteria to consider a study as principal? Authors should explain this classification and revise it on the manuscript;
- The studies about humans and animals are mixed; it should exist a logic criterion to present these studies and they should be presented in a uniform way; revise, re-write and improve;
- Correct the designation ROS, as ahead in the text, NO is referred and it is a RNS;
- Some of the ideas of the conclusion are not clear along the manuscript, as, once more, the studies are presented and not discussed; revise and improve;
- In the conclusion section a final phrase is missing;
- References are up to date; however, due to the number of studies published about this topic, it should be improved. Some references that are missing: doi: 10.1007/978-3-319-24514-0_11; doi: 10.1111/bph.14801; doi: 10.1038/s41430-018-0309-5; doi: 10.3390/diseases6030073; doi: 10.3390/molecules23071684.
Author Response
Response to Reviewer 1 Comments
Point 1: The abstract should be revised and the sentences shortened (e.g. lines 13/17);
Response 1: As suggested by the reviewer the mentioned sentences in the abstract have been shortened.
“Despite epidemiologic studies indicate that excessive alcohol intake is associated with the development of chronic diseases and other serious problems, an ever-expanding amount of scientific evidence supports an inverse relationship between moderate alcohol consumption, particularly for red wine, and the risk of coronary heart disease (CHD).” was changed as “Benefits from moderate alcohol consumption have been widely support by scientific literature and, in this line, red wine intake has been related to a lesser risk for coronary heart disease (CHD).”
Point 2: The term “antioxidant ingredients” should be revised and improved all along the manuscript;
Response 2: As suggested by the reviewer we changed the term “antioxidant ingredients” to “antioxidant compounds” for all the manuscript.
Point 3: Correct “Trans-Resveratrol” at the beginning of the sentence;
Response 3: As suggested by the reviewer Trans-Resveratrol has been replaced by Trans-resveratrol.
Point 4: The methodology used for the bibliographic search and the criteria for the articles selection should appear at the end of the introduction, before the results;
Response 4: Due to the general approach chosen for this review, all studies regarding red wine and cardiovascular disease published over the last decade have been taken into consideration. As suggested, the authors have added this part at the end of the introduction.
Point 5: In section 2, sometimes the compounds levels are referred and other times not; as so, it should be uniform, the levels should always be referred;
Response 5: As suggested by the reviewer the range of concentration value has been added in the manuscript as well as references.
Point 6: Insert reference in section 2.2.3;
Response 6: As indicated, the references were added in section 2.2.3.
Point 7: Taking the section “3. Factors influencing phenolic content and composition of wine”, the non-phenolic content is not influenced by external factors?”; authors should clarify this question and add this topic in the manuscript;
Response 7: As rightly suggested by reviewer we have changed the title of this section from "Factors influencing phenolic content and composition of wine" to "Factors influencing phenolic bioactive compounds and composition of wine" give that also the no-phenolic compounds are influenced by external factors.
Point 8: The manuscript is very descriptive and should discuss more the ideas and the results of the studies cited: Authors should improve this aspect;
Response 8: As rightly suggested by the reviewer, we proceeded to add the suggested information to the manuscript.
Point 9: What authors mean when they write "Principal studies". What was the criteria to consider a study as principal? Authors should explain this classification and revise it on the manuscript;
Response 9: With the term "Principal studies", the authors mean the main relevant scientific researches available in the last decade in which were reported useful data to compare alcohol consumption, especially red wine, with different putative mechanisms of action. As suggested, the authors considered more correct to change the term "Principal studies" as "Summary of studies".
Point 10: The studies about humans and animals are mixed; it should exist a logic criterion to present these studies and they should be presented in a uniform way; revise, re-write and improve;
Response 10: As suggested by the reviewer, we have revised this section and improved the presentation mode.
Point 11: Correct the designation ROS, as ahead in the text, NO is referred and it is a RNS;
Response 11: As rightly suggested by the reviewer, we proceeded to change the designation ROS as RNS.
Point 12: Some of the ideas of the conclusion are not clear along the manuscript, as, once more, the studies are presented and not discussed; revise and improve. In the conclusion section a final phrase is missing;
Response 12: As required by the reviewer, the studies reported in the discussion section were improved for helping the comprehension of the conclusion. Furthermore, a final sentence was added in the conclusion section.
Point 13: References are up to date; however, due to the number of studies published about this topic, it should be improved. Some references that are missing: doi: 10.1007/978-3-319-24514-0_11; doi: 10.1111/bph.14801; doi: 10.1038/s41430-018-0309-5; doi: 10.3390/diseases6030073; doi: 10.3390/molecules23071684.
Response 13: As suggested by the reviewer, we have added in the manuscript the provided references.
Authors thanks the reviewer comments!
Round 2
Reviewer 1 Report
This article was revised appropriately.
I recommend accept.